# Annual Wildflower Strips as a Tool for Enhancing Functional Biodiversity in Rye Fields in an Organic Cultivation System

**Krzysztof Kujawa** [1,*], **Zdzisław Bernacki** [1], **Jolanta Kowalska** [2] 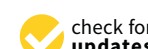, **Anna Kujawa** [1], **Maria Oleszczuk** [1], **Paweł Sienkiewicz** [3] **and Dariusz Sobczyk** [1]

[1] Institute for Agricultural and Forest Environment, Polish Academy of Sciences, Bukowska 19, 60-809 Poznań, Poland; zdzislaw57@gazeta.pl (Z.B.); annakuja@poczta.onet.pl (A.K.); maria.oleszczuk@isrl.poznan.pl (M.O.); daref@poczta.onet.pl (D.S.)

[2] Department of Organic Agriculture and Environmental Protection, Institute of Plant Protection—National Research Institute, Władysława Węgorka 20, 60-318 Poznań, Poland; j.kowalska@iorpib.poznan.pl

[3] Department of Entomology and Environmental Protection, Poznań University of Life Sciences, Wojska Polskiego 28, 60-637 Poznań, Poland; pawel.sienkiewicz@up.poznan.pl

* Correspondence: krzysztof.kujawa@isrl.poznan.pl

**Abstract:** Ecological intensification of agriculture (e.g., with the use of wildflower strips) is being currently discussed as a mean for gaining high yields, preserving high biodiversity of farmland. The aim of this study was to assess the efficiency of annual wildflower strips (WFSs) established in rye field (RF) in (1) increasing species richness and abundance in terms of beneficial arthropod groups (carabids, ground spiders, plant spiders, butterflies, insect pollinators and plant-dwelling insect predators), (2) decreasing the abundance of insect pests, (3) decreasing damages of the crop, and (4) increasing the yield. The field survey was carried out in 2019, in two WFSs and in the adjacent crop field at the distances of 3, 9, 21 and 45 m. The study was not skewed by pesticide use as it was carried out on an organic farm. Mean "site" species numbers ($\alpha$-diversity) and the abundance of most groups were found to be significantly higher in WFSs than in RF. A negative relationship was found in most groups between distance from WFSs and species numbers and abundance. The mean total abundance of all observed pest insects was positively related to distance from WFSs and increased by 83% at distances between 3 and 45 m from WFSs. There was a negative exponential relationship between aphid abundance and total predator abundance, which suggests a mechanism reducing aphid abundance resulting from high levels of predator abundance in the nearby WFSs. The study shows that annual WFSs can be an efficient measure for enhancing cropland biodiversity and should be taken into account in agri-environmental schemes in the Common Agricultural Policy after 2020.

**Keywords:** ecological intensification; taxonomic diversity; life traits diversity; pest control; pollinators

## 1. Introduction

The intensification of agriculture and the conversion of natural or semi-natural habitats to farmed land has led to a rapid decline in the biodiversity of farming areas [1], and hence also of predatory and parasitoid species, which are important agents of biological pest control. In consequence, the application of pesticides has intensified, themselves bringing about a variety of adverse environmental side effects [2].

There is thus an urgent need for solutions that can mitigate the environmental stress associated with intensive agriculture. One of the most important measures is diversification of the agricultural landscape structure [3], which is widely recognized as a prerequisite for achieving a balance between highly efficient farming and high levels of biodiversity [4], while ensuring that ecosystem services remain at an appropriate level [5]. Field margins are among the crucial elements of diverse farmland [6], and they

enable ecological intensification of agriculture which is based on an increase in the intensity of ecological processes supporting production, such as biological pest control, pollination or nutrient cycling [7].

One of the measures contributing to ecological intensification is the establishment of wildflower strips (WFSs), which significantly increase the functional diversity of invertebrates in arable fields, including pollinators [8], parasitoids, predatory insects and spiders [9]. The last three mentioned arthropod groups can significantly increase the efficiency of biological pest control [10]. Another reason for establishing WFSs is the need to prevent the loss of biodiversity in general, including that of insects, whose resources are rapidly declining [11]. Although WFSs are already very common in some countries (Switzerland, the Netherlands and Germany), they are not used in other countries with well-developed agriculture, such as Poland.

It is usually recommended that WFSs should be used as permanent landscape elements, which are more efficient than annual ones [12]. That is for two reasons: firstly, they provide an over-wintering habitat for a variety of arthropods, and secondly, species richness and diversity is expected to increase with advancing age according to the theory of ecological succession. However, annual strips also have advantages. For example, they can easily be established from year to year in different places, and they do not require any special care. Annual WFSs could be the first step towards introducing WFS measures in regions where they have not been used before or as a supplementary tool on farms where perennial WFSs cannot be used.

Several studies have investigated the effects of annual WFSs on crop field biodiversity in Europe, e.g., in maize in Germany [13], in cereals in Belgium [14], in melon crops in Spain [15] and in red clover in Sweden [16]. These studies show that establishing annual WFSs can be effective means for increasing both taxonomical and functional diversity of insects in crop fields. However, their efficiency seems to be related to landscape heterogeneity (the higher heterogeneity, the higher efficiency) [14,16]. The overwintering of insects in WFSs (including annual ones) has been studied in winter wheat in Switzerland [17]. In the paper, the authors underline that annual WFSs are poor overwintering habitats for arthropods, and they may act as ecological traps, when they are ploughed during the overwintering period.

If WFSs are to be maximally effective, one must know the extent of a WFS's impact on the adjacent crop field, and recommendations regarding the arrangement of such strips should include information about the optimal distance between them. To date, the relationship between the distance from a WFS and the magnitude of a WFS's impact on cultivated fields has only rarely been studied—in oilseed rape [13], cereal [14], cabbage [18], strawberry [19], and blueberry fields [20]. The effect of the distance from WFSs on arthropods was assessed in these studies with the use of a different approaches (various distance ranges), and the results are ambiguous.

The aim of this study was to examine the effect of annual WFSs on (i) the species richness and abundance of insects and spiders, including beneficial functional groups (pollinators and predators); (ii) pest abundance; and (iii) rye yield and damage to rye ears. A key part of the study design was assessing the effect of distance from the WFSs. To compensate for the small number of replications (two WFSs, one cultivated field), a multi-taxa approach was applied. Four taxonomical and /or functional groups of insect species, and two functional groups of spider species were studied.

The study was carried out on an organic farm that did not use pesticides. This enabled us to detect a pattern of species richness and abundance of invertebrates that was not biased by pesticide use, which is probably only rarely possible in a typical, intensively-managed crop field. Among the papers cited above, only Sigsgaard et al. [19] provided information on the pesticides used (pyrethroids were applied on one of the three farms they studied).

## 2. Material and Methods

The study was carried out in 2019, in a rye field belonging to the organic Juchowo Farm, situated in the post-glacial, heterogeneous landscape of north-western Poland (N 53.697143, E 16.507382), that consists of various forest patches, cultivated crop fields and grasslands, as well as lakes and other water bodies.

Two sample areas (Sectors A and B) were established in this field. The field survey was conducted in one WFS per sector, and in adjacent rye field. The sampling design in both sectors was identical.

## 2.1. Experimental Wildflower Strips

Two annual wildflower strips (hereafter referred to as WFS-A and WFS-B) were prepared in Sectors A and B at the beginning of May, 90 m apart. They were $300 \times 6$ and $200 \times 6$ m in size, respectively. A mixture of 13 plant species was used (the seed-mass in g/100 m$^2$ and its percentage in seed mixture are given in parentheses), as follows: *Fagopyrum esculentum* (47.10, 59.0%), *Calendula officinalis* (7.77, 9.7%), *Agrostemma githago* (5.98, 7.5%), *Centaurea cyanus* (4.78, 6.0%), *Tagetes tenuifolia* (4.20, 5.3%), *Matthiola longipetala* (2.52, 3.1%), *Chamomilla recutita* (1.67, 2.1%), *Papaver rhoeas* (1.43, 1.8%), *Valerianella locusta* (1.20, 1.5%), *Consolida regalis* (1.08, 1.3%), *Nigella arvensis* (1.08, 1.3%), *Anethum graveolens* (0.48, 0.6%) and *Satureja hortensis* (0.48, 0.6%). As there is no experience of applying WFSs in Poland, the composition of the plant species mixture was established on the basis of recommendations from the Swiss Federal Research Station for Agroecology and Agriculture [21]. However, the original Swiss seed mixture was modified. Firstly, the relative amount of *F. esculentum* was lowered by 40% (from 78.50 to 47.10 g/100 m$^2$) to reduce its dominance, and to provide more space for other plant species. Secondly, some species had to be removed to comply with Polish national regulations on organic agriculture, or because they do not occur in Poland; they were replaced with other, similar species. Ultimately, six of the species were the same as in the Swiss mixture, while the other seven were intended to be functional substitutes for the Swiss species.

## 2.2. Sampling Design

As the sampling design (Figure 1) in both Sectors (A and B) was the same, the description below (number of transects, sections, traps, sample plots, field survey dates) relates to a single sector. To assess the effect of distance from the WFSs on the species richness and abundance of the arthropod groups studied and the rye yield, four transects were laid out parallel to the WFSs. The distance between each of the four transects and the WFS edge was 3, 9, 21 and 45 m respectively, i.e., the transects were 6, 12 and 24 m apart. Transects A4 and B4 (45 m from the WFSs) were treated as controls representing the rye field and assumed to be wholly unaffected by the WFSs. Seven sections, each 10 m in length, were established along the WFS and each field transect. The sections were spaced at intervals of 10 m (i.e., within about 150 m long parts of the WFSs).

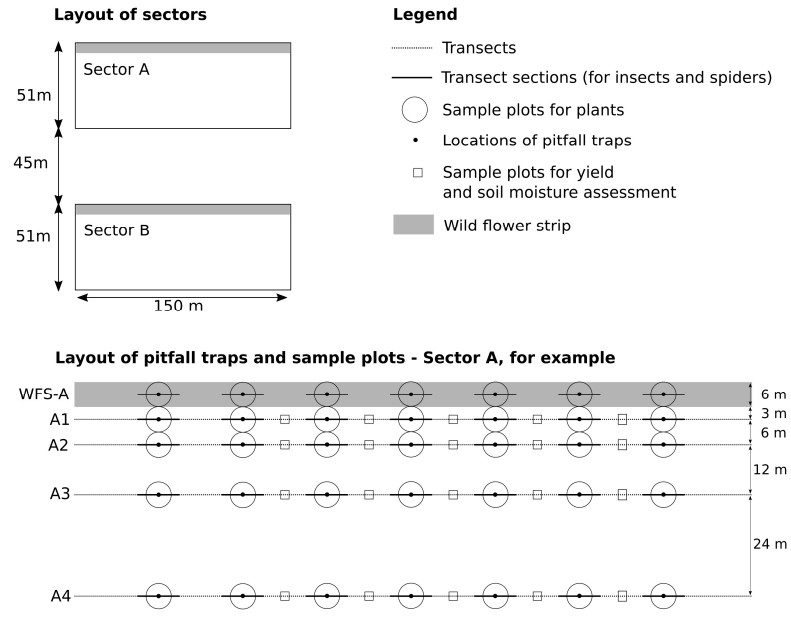

**Figure 1.** Layout of sectors and transects, and sampling design. WFS-A—wildflower strip, A1-A4—transects in the rye field.

### 2.3. Field Survey Method

The field survey was conducted from 20 May to 2 September, the various aspects of the survey (see points A–H below) being carried out with different frequencies.

A. The flora was assessed using phytosociological relevés [22]. Plant species richness and individual species abundance were assessed four times (20 May, 27 June, 15 July, 2 September) in 35 circular plots per sector (7 in the WFSs and 28 in the four field transects), i.e., within 3 m of the central points of the transect sections. All the plant species were recorded, and their abundance was quantified using a six-point scale (the ranges refer to the percentage cover), with 5 representing 76–100%; 4 being 51–75%; 3 being 26–50%; 2 being 11–25%; 1 being 1–10%; and 0.1 for values < 1%. The averaged species abundance during the peak of vegetation development (June–July) was used in the analyses.

B. The activity densities of carabid beetles and ground-dwelling spiders ("ground spiders") were assessed using pitfall traps (height—13 cm, and diameter—8.5 cm, filled with 75% glycol and a few drops of detergent), fitted with a plastic roof. They were installed at the central points of all the WFS sections and the field transect sections (35 traps per Sector, Figure 1) from 20 May to 2 September. The traps were emptied seven times, every 14 days on average. All the individuals have been identified to species-level. All the spiders collected in the pitfall traps were treated as "ground spiders".

C. Plant-dwelling predatory insects (hereafter referred to as "predators"), pollinating species ("pollinators") other than butterflies and plant-dwelling spiders ("plant spiders") were collected four times in all the WFS sections and all the field transect sections, every two weeks starting from 20 June and ending on 15 July, using entomological sweep-nets (10 sweeps per section); all the spiders captured were treated as "plant spiders". The "predators" represented four families: Coccinellidae (Coleoptera), Vespidae (Hymenoptera), Chrysopidae (Neuroptera) and Asilidae (Diptera); and "pollinators" represented two families: Apidae and Bombycidae (Hymenoptera).

D. Butterflies were counted using the transect method [23], walking along all the sample sections in the WFSs and rye, every two weeks starting from 3 June and ending on 15 July. The sections were walked slowly (1–2 km per hour), and all the adult butterflies were counted in an imaginary box 2.5 m to each side and 5 m in front and above the observer [24]. The butterflies were identified during the field survey.

E. Pest insects belonged to the orders Hemiptera, Coleoptera and Thysanoptera, i.e., the most important pests of cereal crops. They were counted and identified directly in the field. All the specimens were recorded by scouting 25 randomly selected plants in each field transect section (28 sections per Sector, Figure 1) on 20 May, 8 June and 2 July (growth stages BBCH: 56, 69, 87, respectively). In July, Aphidoidea specimens were counted on 10 randomly selected rye ears per field transect section twice a month [25].

F. Ear damage frequency (DAM) was assessed on 1 of July on the basis of 10 randomly selected ears per transect section. All kind of ear damage (structural or colour changes, malformation, absence of grains, physical damage to grains etc.) were noted, and the number of damaged ears per 10 ears were used as a measure of the frequency of ear damage. The damage was caused mainly by *Aelia acuminata* (Heteroptera: Pentatomidae) and *Eurygaster maura* (Heteroptera: Scutelleridae).

G. The rye yield was estimated in July, a few days before the harvest, by measuring the grain/ear numbers in 100 randomly selected ears/transect ($x$), mean grain weight ($y$), and rye shoot density in five 0.25 m$^2$ plots per transect (Figure 1) ($z$). The yield per plot was calculated according to the formula $x \times y \times z$ and extrapolated to one hectare.

H. The soil moisture content was measured as it is a key factor in vegetation development and crop plant yields. It was estimated using the weighing and drying method. Five soil samples per field transect were taken at a depth of 10 cm using 100 cm$^3$ Kopecky cylinders in July, just before the harvest. They were weighed to an accuracy of 0.01 g and dried in the laboratory for 48 h at a temperature of 80 °C, after which they were reweighed.

*2.4. Data Presentation and Statistical Analysis*

In the analysis of the spatial variation of arthropod species numbers and abundance the data collected in the second half of June and the first half of July were included (growth stages BBCH 69–89).

The indicator Chao2 [26] was used for comparison of species richness between the WFSs and the rye field, while mean number of species per section or trap was used in the analysis of spatial variation of arthropod occurrence in relation to the distance from the WFSs.

The species richness (Chao2) of the taxa was estimated using the "wiqid" package [27] in the R environment [28]. Incidence data for all the dates were used for 14 sections (or traps) in both WFSs, and for 14 sections (or traps) located in the "control" field transects (A4 and B4). "Predators" and "pollinators" were excluded from this analysis as they were not identified to species level. The significance of the differences in Chao2 was determined by a comparison of the confidence intervals (95%CI). The difference has been interpreted as significant ($p < 0.05$), when the CIs did not overlap.

An index of the total abundance of predatory arthropods was used in the analysis of the impact of WFS distance on aphid abundance. This was the sum of the standardized densities (or activity densities) of carabids, "ground spiders", "plant spiders" and "predators", increased by the smallest of these values, and additionally by 0.1 to give positive values (necessary to use the logarithmic function in the next stage of the analysis). The curve illustrating the relationship between aphids and predator abundance was fitted using the model $y = a \cdot ln(x) + b$, with the use of Statistica [29].

The mean abundance of the insect pests occurring in the rye field nearby and far from the WFSs (3 m vs. 45 m) was compared with the use of the data from all the sample sections (from the Sectors A and B).

The statistical significance of the effects of the predictors (habitat variables) was verified using a general linear model (GLM) for arthropod abundance or activity density, vegetation cover, rye yield and soil moisture content. A generalized linear model (GLZ) was used for species numbers and ear damage frequency. As the data were not over-dispersed ($p > 0.1$ in the over-dispersion test), the Poisson error distribution was used in GLZ. Over-dispersion was assessed using the "AER" R package [30]. When referring to the F-value (the results of using GLM), the degree of freedom and sample size are given in lower case.

Three predictors were used in the analyses:

(a) habitat type (HAB)—a categorical fixed factor with two levels: WFS ("strips") and crop field ("field") which was represented by the transects A4 and B4 only, as the most distant from the WFSs;
(b) distance (DIST) from WFSs—a numerical variable with four values (3, 9, 21 and 45 m);
(c) sector of the study area (SECT)—a categorical fixed factor with two levels ("A" and "B").

The SECT has been defined as fixed factor because the analyses of plant communities had shown significant differences in weed species richness and abundance (see Section 3.2), that are one of the key driver shaping insect and spider assemblages in crop fields [31]. Level "A" and "B" represented high and low richness and abundance of weeds, respectively. Two interactions were considered—HAB × SECT and DIST × SECT.

Wherever the arithmetical means are reported, 95% confidence intervals (CI) or standard deviations (SD) are given.

## 3. Results

The analyses were based on 210 (10 transects × 7 plots × 3 dates) phytosociological relevés (lists of plant species with their abundance), 419 specimens of pest insects observed, 15,239 carabid beetles caught, 227 butterflies sighted, 929 plant-dwelling insects observed (other than carabids and butterflies), 9578 ground-dwelling spiders caught and 328 plant-dwelling spiders caught. In addition, 560 rye ears were assessed for grain damage (2 sectors × 4 transects × 7 sections × 10 ears), 800 ears (8 transects × 100 ears) for rye-yield assessment, and the number of shoots from 40 sample plots (8 transects × 5 plots) for rye density assessment.

### 3.1. Soil Moisture

The mean soil moisture content in Sector A was 14.5% ± 0.9% (CI); this figure was significantly higher ($F_{1,40}$ = 54.7, $p < 0.001$) than in Sector B, where it was 9.1% ± 0.9% (CI). The effect of DIST was insignificant ($F_{1,40}$ = 0.74, $p = 0.39$). General statistics for the model were the following: $F_{1,40}$ = 27.7, adjusted $R^2$ = 0.59 and $p < 0.001$.

### 3.2. Vegetation Cover in the Wildflower Strips and Crop Field

As many as 100 plant species were found in the study area, i.e., 72 species in the WFSs and 70 in the rye field. As expected, *F. esculentum* was dominant (40–50% cover), while the cover of *Chenopodium album*, *Polygonum persicaria*, *Thlaspi arvense*, *Secale cereale*, *C. cyanus* and *C. recutita* was also large. Only the last two species were included in the seed mixture; the others were grown from the soil seed bank. Eleven of the 13 species sown appeared, while 61 other species grew spontaneously (see Section 3.2 for more details).

The summarized cover of the twelve most abundant species in the WFSs made up 80% of the total vegetation cover (Table S1). The number of species per plot (averaged for June and July) was significantly correlated with the averaged vegetation cover in both the WFSs ($n = 14$, $r = 0.60$, $p = 0.022$) and the rye field ($n = 56$, $r = 0.54$, $p < 0.001$) (Figure S1), so it was considered a proxy for plant cover and plant diversity, and further analyses included the mean number of species per plot only. This was 28.4 ± 1.2 (CI) in the WFSs, which was higher ($F_{1,70}$ = 384.4, $p < 0.001$) than in the rye field, where it was 15.9 ± 0.6 (CI). The number of plant species per plot was higher ($F_{1,70}$ = 36.1, $p < 0.001$) in Sector A than in Sector B by an average of ca 4 species per plot in both habitats. The SECT × HAB interaction was statistically insignificant ($F_{1,70}$ = 0.36, $p = 0.78$). The analysed predictors (SECT, HAB, SECT × HAB) explained 86% of the variability in the number of plant species and the relationship was highly significant ($F_{1,70}$ = 145.9, $p < 0.001$).

### 3.3. Species Numbers, Activity Density and Abundance of Arthropods in the Wildflower Strips

As many as 46 species of carabid beetles (Table S2) were trapped in the WFSs. All the species feed on invertebrates, and 36% of the individuals from 21 species were specialized zoophages, which can play an important role in reducing the number of crop pests. The most abundant species was *Harpalus rufipes* (hemizoophage), which made up 45.8% of the numbers of individual beetles. The most abundant of the 47 "ground spider" species (Table S3) was *Pardosa agrestis* (24% of all individuals), a relatively large-bodied, actively hunting species. The majority of individuals in the assemblage (83%) were non-web-building species (76% of all species). Twenty species of "plant spiders" were caught (Table S4): the majority of individuals (76%) here belonged to 19 web-building species (95% of all species). There were 14 species of butterflies from five families. The most abundant species was *Vanessa cardui* (54%), and the proportion of individuals of this and two other highly abundant species (*Pieris napi* and *P. rapae*) was 71% (Table S5). The respective abundances of "predators" and "pollinators" were 4.2 ± 1.5 (SD) and 1.6 ± 0.7 (SD) ind./section/visit. The coefficient of variability of the taxa abundance ranged from 0.2 to 0.5 (Table S6).

### 3.4. Total Species Richness, Species Numbers per Plot and Mean Abundance (or Activity Density) of Arthropods—Comparison between the Wildflower Strips and the Crop Field

Species richness (Chao2) between the WFSs and the rye field differed markedly only with respect to the "ground spiders" (Figure 2A). In contrast to the similarity between the WFSs and the rye field in species richness for most of the taxa compared, the species numbers per section (or trap) in the taxa were significantly higher (marginally so for ground spiders) in the WFSs (Figure 2B, Table 1). SECT was a significant predictor for "ground spiders": their mean species number per section in Sectors A and B was 7.3 and 5.1, respectively, the difference being statistically significant (Table 1). There was SECT × HAB interaction in the case of "plant spiders" (Table 1), reflected by the large difference between the WFS and the rye field in Sector A (Figure 2B) but no such difference in Sector B.

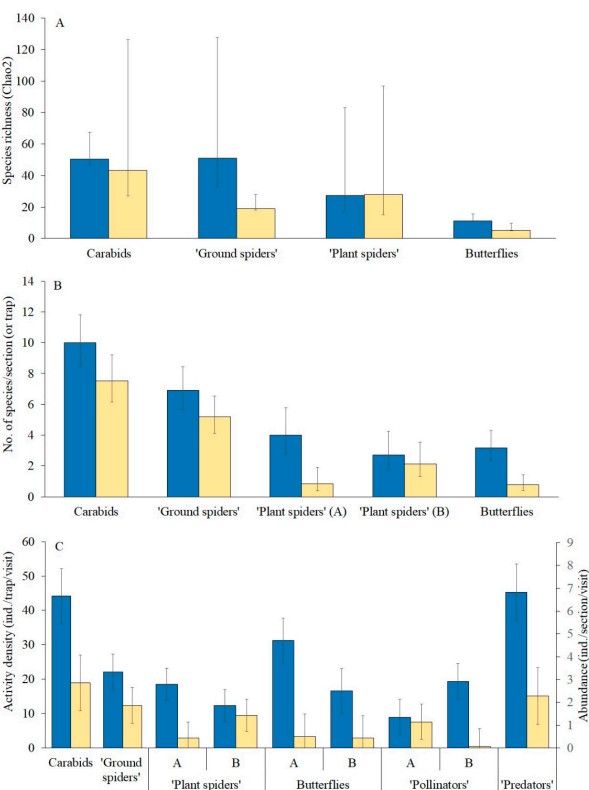

**Figure 2.** Species richness (Chao2) (**A**), mean species number (**B**) and mean abundance (**C**) per section (or trap) for the arthropods in the wildflower strips (blue bars) and the rye field (yellow bars) in June–July. In the bottom plot, the activity densities of carabids and "ground spiders" are given on the left-hand Y-axis, the abundance of the other groups on the right-hand Y-axis. The model presented in Table 1 was used to calculate the 95% CIs. Sectors A and B are shown only for those groups for which a significant Habitat × Sector interaction was demonstrated (Table 1).

**Table 1.** Effect of predictor sets: (1) "Habitat" (H) and "Sector" (S), and (2) "Distance" (D) and "Sector" (S), on arthropod species richness per section or per trap (in the GLZ model with Wald's statistics given), and on mean abundance per section or on activity density per trap (GLM model with F statistics given). The intercept is not shown. W—Wald's statistics, F—Fisher statistics, *p*—statistical significance. The predictors significant at *p* < 0.05 are marked bold.

| Expl. Variable | Predictor Sets | Predictors | Carabids | | Ground Spiders | | Plant Spiders | | Butterflies | | Predators | | Pollinators | |
|---|---|---|---|---|---|---|---|---|---|---|---|---|---|---|
| | | | W/F | *p* | W/F | *p* | W/F | *p* | W/F | *p* | F | *p* | F | *p* |
| Species richness | (1) | H | **4.6** | **0.032** | 3.3 | 0.068 | **9.8** | **0.002** | **17.0** | **0.000** | | | | |
| | | S | 1.0 | 0.319 | **4.6** | **0.031** | 0.9 | 0.351 | 2.0 | 0.154 | | | | |
| | | HxS | 1.5 | 0.226 | 0.9 | 0.355 | **5.3** | **0.021** | 0.8 | 0.373 | | | | |
| | (2) | D | **3.7** | **0.05** | **10.6** | **0.001** | 0.4 | 0.554 | 1.5 | 0.219 | | | | |
| | | S | 0.0 | 0.998 | 0.1 | 0.714 | 0.6 | 0.437 | **11.9** | **0.001** | | | | |
| | | DxS | 0.5 | 0.500 | 0.5 | 0.489 | 0.8 | 0.367 | **6.6** | **0.010** | | | | |
| Abundance or activity density | (1) | H | **21.0** | **0.000** | **7.3** | **0.012** | **16.5** | **0.000** | **43.5** | **0.000** | **32.7** | **0.000** | **16.8** | **0.000** |
| | | S | **12.2** | **0.002** | **12.9** | **0.000** | 0.0 | 0.918 | **5.8** | **0.025** | 0.3 | 0.608 | 0.4 | 0.512 |
| | | HxS | 0.5 | 0.498 | 0.5 | 0.482 | **7.9** | **0.010** | **5.1** | **0.034** | 3.6 | 0.069 | **12.4** | **0.002** |
| | (2) | D | 1.4 | 0.237 | **20.0** | **0.000** | 0.1 | 0.767 | 0.0 | 0.883 | 3.7 | 0.060 | **7.4** | **0.009** |
| | | S | **16.6** | **0.001** | **7.6** | **0.008** | 1.7 | 0.203 | **10.7** | **0.002** | **30.7** | **0.000** | **38.1** | **0.000** |
| | | DxS | 0.0 | 0.954 | 0.3 | 0.611 | 0.6 | 0.425 | 3.3 | 0.073 | 2.0 | 0.159 | **4.5** | **0.038** |

The abundance (or activity density) of all the groups of species was significantly greater in the WFSs than in the rye field (Figure 2C, Table 1). A significant Sector × Habitat interaction was found in four

of these groups. The differences between the WFS and rye field with regard to "plant spider" and butterfly abundance were significantly larger in Sector A, in contrast to "pollinator" abundance, which did not differ between the WFS and rye field in Sector A, but did differ strongly and significantly in Sector B (Figure 2C).

*3.5. Effect of Distance from a Wildflower Strip on the Species Richness and Abundance (or Activity Density) of Beneficial Insects and Spiders*

The expected negative effect of DIST on arthropod species numbers and abundance (or activity densities) in the rye field was found in most of the groups (Figures 3 and 4), but it was only statistically significant for the species numbers of carabid beetles and "ground spiders", and for the abundance of "ground spiders", "pollinators", and "predators" (marginal significance $p = 0.06$) (Table 1). There were two significant ($p < 0.05$) or marginally significant ($p < 0.1$) interactions between the sectors and distance (SECT × DIST) (Table 1). The butterfly species number and abundance were significantly higher in Sector A, and their relation to DIST was detectable there, in contrast to Sector B, where the abundance of butterflies was very low and there was no relation to DIST (Figure 3). A similar pattern was found with respect to predator abundance (Figure 4), which was significantly (4-fold) higher in Sector A, where it tended to decrease with increasing DIST. Predator abundance in Sector B was not related to DIST. Pollinator abundance was also much larger in Sector A, but no relation to DIST was found in either Sector.

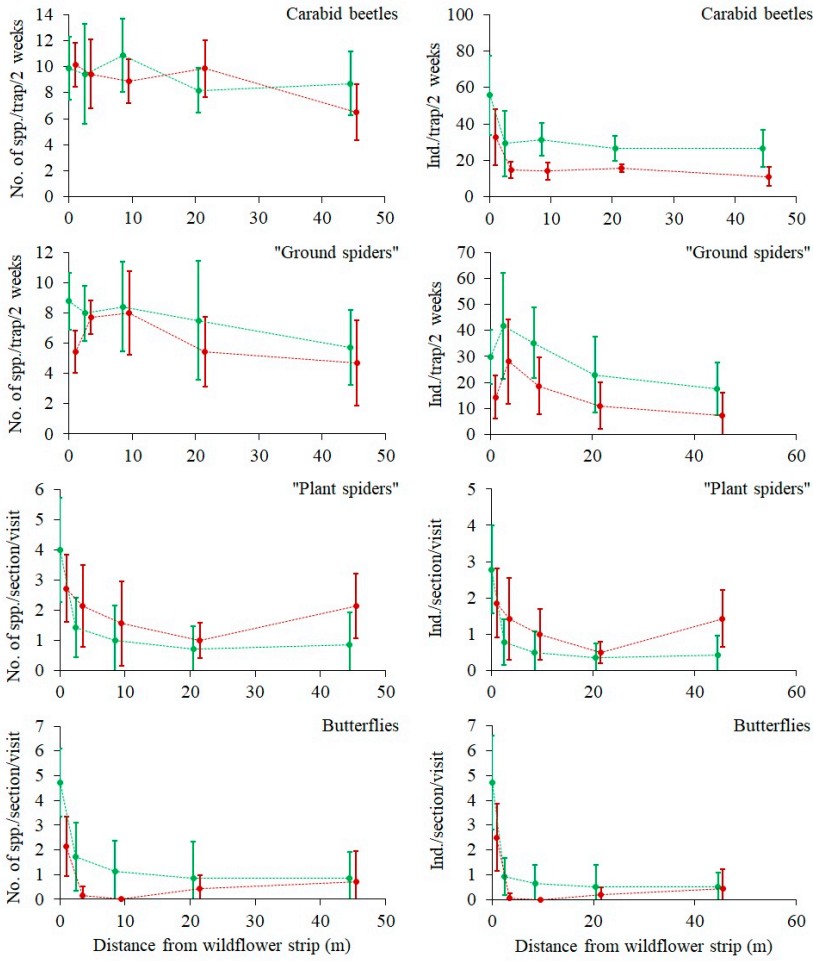

**Figure 3.** Species number per section or per trap (left-hand column) and mean abundance per section (or activity density per trap) (right-hand column) of carabid beetles, "ground spiders", "plant spiders" and butterflies in the wildflower strips (distance—0 m) and for distances of 3, 9, 21 and 45 m from the WFSs. Sector A—green, Sector B—red. The standard deviations are shown.

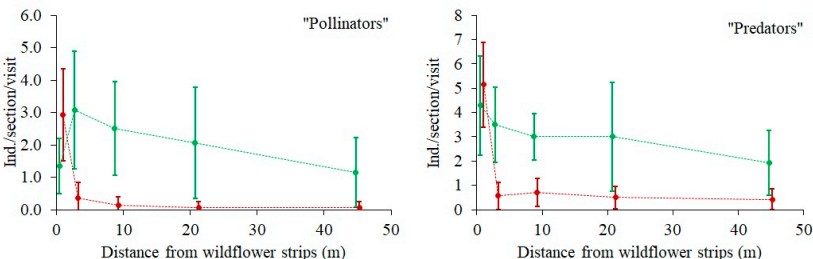

**Figure 4.** Mean abundance of pollinators and predators in the wildflower strips (distance—0 m) and for distances of 3, 9, 21 and 45 m from the WFSs. Sector A—green, Sector B—red. The standard deviations are shown.

### 3.6. Effect of the Wildflower Strips on the Occurrence of Pest Insects

The mean total abundance of all observed pest insects, i.e., *Aelia acuminata* (Hemiptera: Pentatomidae), Thripidae (Thysanoptera), *Oulema melanopa*, *Oulema gallaeciana* (Coleoptera: Chrysomelidae), *Sitobion avenae*, *Macrosteles laevis*, *Rhopalosiphum padi*, *Metopolophium dirhodum* (Hemiptera: Aphididae), *Eurygaster maura* (Hemiptera: Scutelleridae), *Phyllopertha horticola* (Coleoptera: Scarabaeidae), was positively related to DIST. It increased from 2.8 ± 1.0 (SD) ind./25 plant/visit at 3 m from the WFSs to 5.5 ± 4.3 (SD) ind./25 plant/visit at 45 m from the WFSs. In the case of Aphidoidea only, the relationship was the same in both sectors (Figure 5). There was a logarithmic relationship (R = 0.4) between aphid abundance and total predator abundance (Figure 6).

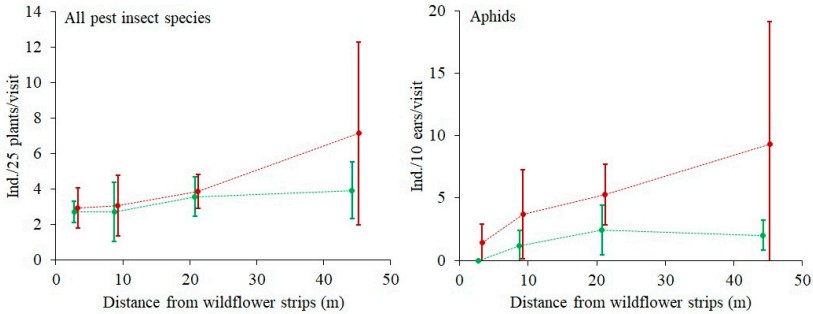

**Figure 5.** Total abundance of pest insects (**left**) and aphids only (**right**) at distances of 3, 9, 21 and 45 m from the annual wildflower strips in Sectors A (**green**) and B (**brown**). The 95% CI is shown.

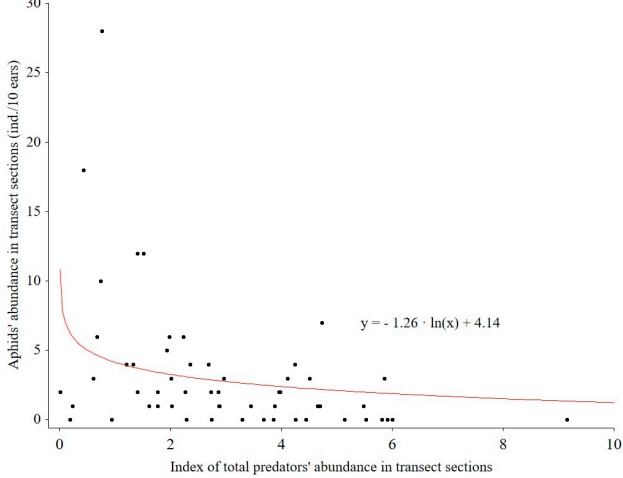

**Figure 6.** Relationship between aphid abundance and the index of the total abundance of predators (insects and spiders).

### 3.7. Ear Damage

The numbers of damaged ears (N = 193) in the sectors ranged from 0 to 7 per 10 ears, the mean being 3.2 ± 1.7 (SD) and the median 3. Most of the damage (90%) was caused by *A. acuminata*. The effect of DIST on DAM was on average insignificant (Wald = 2.2, $p$ = 0.139). However, there was a significant SECT × DIST interaction (Wald = 5.4, $p$ = 0.020). The pattern of relationships between DIST and DAM differed between Sectors A and B. DAM was not related to DIST at all in Sector B, whereas a positive correlation was found in Sector A. This was demonstrated by the Gamma correlation coefficient, which amounted to -0.13 ($p$ = 0.51) and 0.52 ($p$ = 0.004), respectively.

### 3.8. Yield

The difference between Sectors A and B in the mean rye yield, i.e., 4163 ± 204 (CI) kg/ha and 4530 ± 203 (CI) kg/ha, respectively, was statistically significant ($F_{1,792}$ = 6.2, $p$ = 0.012). The yield varied strongly between the distances ($F_{3,792}$ = 22.2, $p$ < 0.001). The pattern of these differences varied between the sectors (significant SECT × DIST interaction, $F_{3,792}$ = 49.0, $p$ < 0.001): the yield in Sector B tended to decrease with increasing DIST, from 5917 to 2842 kg/ha on average, whereas there was no such dependence in Sector A. The mean yields for the consecutive distances (3, 9, 21, 45 m) in Sector A were 3153; 4948; 3456 and 5094 kg/ha.

## 4. Discussion

The importance of perennial WFSs for increasing agricultural biodiversity has been documented in many papers (see [31] for a review). It has recently been recommended that wildflower strips should again be incorporated in the agri-environmental schemes of the European Union [8,32].

The results of our study confirm and expand our knowledge of the importance of annual WFSs for enhancing crop field biodiversity. Firstly, within just three months of being established, they were found to have become a habitat in which the species richness of carabid beetles, spiders and butterflies, and the abundance (or activity density) of all the studied groups, i.e., carabid beetles, spiders, butterflies, other pollinators, as well as predatory plant-dwelling insects, was significantly higher than in the adjacent rye field. This is in line with other studies of annual WFSs, which show that such strips are favourable habitats for many beneficial insects and spiders [13,14,16]. Although annual flower strips develop in a relatively short time (for 3–5 months before harvest), they are much more favourable for many arthropods, when compared to adjacent cultivated fields. This is due to much greater diversity of plant species, a more diverse structure of the habitat (caused by a higher plant diversity) and the lack of chemical plant protection. Due to the rich species composition of nectar-producing plants, flower strips are usually an attractive habitats for many insect species that feed on pollen and flower nectar.

Secondly, the studied annual WFSs turned out to be "exporters of arthropods" to the crop field. The species richness of carabid beetles and ground spiders, and the abundance (or activity density) of ground spiders, "pollinators" and "predators" (marginal significant effect) increased with decreasing distance from the WFS. This result is consistent with the spatial pattern of staphylinid, carabid and spider activity density, and also the species richness observed in the adjacency of annual WFS in rapeseed fields in Germany [13]. A poor distance effect was found in a study of pollinators in melon fields in Spain, where the positive WFS effect on pollinator visit frequency was only detectable at a distance of less than 10 m from the WFS [15]. A few other studies considered just two distances from a WFS. In a study of carabids in cereal fields in Belgium, the effect of distance (2 vs. 30 m.) on total species richness and activity density was found to be insignificant [14]. Furthermore, a study conducted in red clover fields in Sweden mostly found a weak and insignificant effect of distance (8 vs. 100 m.) from a WFS on bee abundance [16]. The only result of this study confirming the positive effects of WFSs on bee abundance in crop fields was in relation to honeybee abundance, which was significantly higher closer to the WFS during the flowering period of both phacelia and clover. In the study on the effect of flowering plantings on blueberry pollination [20] the bee abundance on the blueberry bushes situated few meters from flowering plantings was higher when compared to blueberries growing close to grassy field margins. This study also showed a positive effect of flowering

plantings on the yield of the blueberry bushes growing 15 m from the flowering plantings, however the data on the bee abundance on these bushes was not presented in the paper [20]. Thus, the results presented above are varied and ambiguous. Moreover, any generalization of the results of these studies is hampered by different approaches to measuring the effect of the distance from WFSs (different distances were used, at various spatial resolution), different taxa and crops studied as well as by lacking data on the use of insecticides, which can bias study results. However, extensive study (many sample plots, high spatial resolution of measuring the effect of the distance from WFSs) carried out in rapeseed fields [13] and the results presented in this paper shows, that abundance and diversity of insects and /or spiders decrease gradually together with the distance from WFSs.

Thirdly, we found reciprocal relationships between aphid abundance and the total abundance of predators (insects and spiders together); this was combined with an increase in species abundance of aphids and all pest species with increasing distance from the WFS. These observations confirm that WFSs, even annual ones, can contribute to reducing the number of aphids. This is in line with the results obtained by Hatt et al. [33], who stated that the abundance of aphids (especially winged ones) in wheat growing in between WFSs was lower than in monoculture wheat plots. In our study, however, such relationships were not reflected by the spatial pattern of the rye yield. Aphids are pest insects whose numbers are most often assessed in relation to wildflower strips or other elements of the agricultural landscape. The beneficial impacts of these elements on reducing aphid populations were also confirmed in a study by Toivonen et al. [34]: they concluded that epigeal predators had moved from the wildflower strips to fields before aphid numbers peaked and that the proximity of the grass and wildflower strips reduced aphid pressure during aphid outbreaks. To the best of our knowledge, our study is the only one in Europe where the effects of annual WFSs on cereal-pest abundance has been systematically studied.

In our study, we observed some variation between Sectors A and B as regards the patterns of differences between WFSs and the crop field, and the effect of distance to WFS on the studied taxa. The soil in Sector A had a higher moisture content, which was presumably responsible for the high plant-species richness (and high vegetation cover, see Figure S1) in both the WFSs and the rye field. As vegetation cover and species richness governs the natural resources available to invertebrates, the differences in mean number of species, and their abundance or activity density (usually higher in Sector A, Figure 3) presumably reflected the differences in vegetation cover and species richness between the sectors. These differences between the Sectors A and B are analogous to the recognized interaction between WFS efficiency and the landscape context [35], which affects taxa-dispersion mechanisms and , therefore, both gamma diversity (species richness within the whole landscape) and alpha diversity (species richness within a given field or WFS).

Presumably, the high vegetation species richness in the Sector A also contributed to low pest abundance (Figure 5), even at the distance of 45 m from the WFS, due to the mean number of species and abundance of pest enemies (Figure 3—carabid beetles and ground spiders, and Figure 4—"predators") which were or tended to be higher in the Sector A, when compared to the Sector B.

The positive effects of WFSs on biodiversity and their relatively high efficiency may be weakened by the ecological trap effect. As annual WFSs are usually ploughed up in autumn, they may indeed turn out to be ecological traps for some of the arthropods inhabiting these strips (as shown above) during the WFS growing season from spring to early autumn. Following the crop harvest, moreover, WFSs may become an even more attractive habitat for arthropods. Such an ecological trap effect caused by WFSs was highlighted by Ganser et al. [17], who concluded that WFSs are valuable overwintering habitats for arthropods, but may turn into ecological traps if ploughed up every year. On the other hand, Füglistaller et al. [36] did not find annual WFSs to be ecological traps for wild bees. These contradictory results are presumably due to differences in the plant species composition of WFSs and /or subtle differences in habitat structure, like the proportion of leaf litter and bare soil, which can be a limiting factor for many taxa searching for suitable overwintering sites. Regardless of the study [36] results, to minimize the undesirable effect of the ecological trap caused by applying annual WFSs, the WFSs should be established in cultivated

fields, where ploughing will be done in the spring of the following year. Finally, it is worth underlining, that perennial WFS do not have these drawbacks, and their efficiency as a tool for ecological intensification increases in following years [20,37].

The research, which results are presented and discussed in this article, is a case study. The field survey was carried out only in one crop field, in which two sectors with the same sampling design were established. The aim of the study was to measure the distance of annual flower strips' impact on some ecological processes that undergo in the crop field (including biological pest control). Such an approach has to be based on taking samples at several distances from the WFSs in a single experimental field. Therefore, the low number of replications in this study was sacrificed in favour of more details in terms of: (1) multi-taxa approach, (2) sampling the invertebrates and the plants at four distances from the WFSs, and (3) sampling the invertebrates and the plants many times in vegetation season to collect representative data for a large part of development period of the crop plant.

## 5. Conclusions

The results of our study and the above discussion indicate that annual WFSs can be a valuable and effective measure for enhancing biodiversity as well as mechanisms for the biological control of pests on organic farms, where insecticide use is minimal. They quickly become local refuges for arthropod assemblages with a high α-diversity and total abundance, including those of insect pest enemies, which can contribute to the control of pest numbers. Although the magnitude of a WFS's impact on cultivated fields differed between beneficial taxa, the results of the study suggest that the distance between parallel WFSs should not exceed 60 m. However, the dependence of WFS efficiency on the landscape structure and /or the local species pool (recognized in some previous studies) may be more pronounced in the case of annual WFSs owing to their brief existence. This is because the abundance and species diversity of arthropod taxa in annual WFSs are more dependent on colonization from adjacent areas than in the case of perennial flower strips, which can support many species autonomously. Therefore, we particularly recommend applying annual WFSs not only on organic farms but also on farms situated in a moderately diverse landscape. In regions with conventional farms but no wildflower strips, they could be applied as a first step towards creating a network of permanent WFSs or as a complement to existing networks of linear green-infrastructure elements, such as hedges and permanent WFSs.

**Supplementary Materials:** The following are available online at http://www.mdpi.com/2073-4395/10/11/1696/s1, **Data1**-arthropod_species_richness_WFS_vs_Rye. **Data2**-plant_and_arthropod_abundance. **Data3**-rye yield and shoot density. **Figure S1**: Correlation between the number of plant species per plot (averaged for June and July) and total vegetation cover in the wildflower strips and the rye field. The total vegetation cover is a sum of the individual plant species covers, that were quantified using a six-point scale (the ranges refer to the percentage cover): 5—76–100%; 4—51–75%; 3—26–50%; 2—11–25%; 1—1–10%; and 0.1—for values < 1%, and averaged for June and July. **Table S1:** Species composition of plants in the annual wildflower strips by cover, averaged for June and July. The plant species cover was quantified using a six-point scale (the ranges refer to the percentage cover): 5—76–100%; 4—51–75%; 3—26–50%; 2—11–25%; 1—1–10%; and 0.1—for values < 1%. The highlighted species were in the seed mix used. **Table S2:** Species composition of carabid beetles in the annual wildflower strips by activity density (individuals/trap/14 days), averaged for June–August. See "Methods" for details. Specialist predatory species are shown in bold. **Table S3:** Species composition of ground-dwelling spiders in the annual wildflower strips by activity density (individuals/trap/14 days), averaged for June–August. See "Methods" for details. **Table S4:** Species composition of plant-dwelling spiders in the annual wildflower strips by abundance (number of individuals per 10 m transect sections per visit), averaged for May–July. See "Methods" for details. **Table S5:** Species composition of butterflies in the annual wildflower strips by abundance (number of individuals per 10 m transect section per visit), averaged for June–July. See "Methods" for details. **Table S6:** Variability in the mean number of species and abundance (or activity density) of the studied taxa in annual WFSs per trap or section (N = 14).

**Author Contributions:** Conceptualization—K.K., J.K., P.S., M.O.; Funding Acquisition—K.K., Methodology—K.K., Z.B., J.K., A.K., M.O., P.S., D.S.; Investigation—Z.B., J.K., A.K., M.O., P.S., D.S.; Data curation—K.K., Z.B., J.K., A.K., M.O., P.S., D.S.; Formal analysis—K.K.; Original draft preparation—K.K.; Review and editing of paper—K.K., Z.B., J.K., A.K., M.O., P.S., D.S.; Visualisation—K.K.; Project Administration—K.K., M.O.; Supervision—K.K. All authors have read and agreed to the published version of the manuscript.

**Funding:** This study was supported by the Ministry of Agriculture and Rural Development in Poland (decision No. PJ.re.027.10.2019).

**Acknowledgments:** We would like to thank the staff of the Juchowo Farm for their cooperation in the implementation of this project, and Judyta Konik, Marlena Michalak and Karol Smolarek from the Institute of Agricultural and Forest Environment, Polish Academy of Sciences, for their assistance with the field data collection and data work-up. The text was proofread by Peter Senn.

**Conflicts of Interest:** The authors declare no conflict of interest.

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
