# Peer review of "Annual Wildflower Strips as a Tool for Enhancing Functional Biodiversity in Rye Fields in an Organic Cultivation System"

_agronomy, doi:10.3390/agronomy10111696_

Round 1

Reviewer 1 Report

General comments:

The paper entitled "Annual wildflower strips as a tool for enhancing functional biodiversity in rye fields in an organic cultivation system" presents interesting data on the influence of WFS on the richness and abundance of different beneficial arthropod groups in rye fields. While other studies focus on one or a few beneficial insect groups, the authors performed an exhaustive sampling of many functional groups including carabids, spiders, predators, butterflies, and pollinators, this strength should be better highlighted.

I find some aspects of the results very interesting, however, the content of the introduction and discussion should be improved in order to facilitate a clear picture of the results of the study. In general, relevant references are provided but the authors have not adequately delved into them.

The authors sampling in just in 2 fields one year, so the 7 repetitions considered in each of the two fields can be considered pseudoreplicates, this weakness of the article should be considered and justified in the discussion. Due to the limitation of the number of replicates, the authors have considered each of the field (sector) as a fixed factor, so that the figures, mainly Fig. 4 and 5, reflect the data of both sectors separately and it seems that it is one of the main objectives, however, it is not specified as an aim to be addressed and it hampers the ability to draw any general conclusions. I recommend considering the sector as a random factor. Otherwise, a better justification of why it is interesting to analyze the differences between them should be included.

The identification methodology for all the individuals sampled remains to be detailed. Besides. several statistical analyzes carried out have not been described in the statistical analysis section.

I have some comments or suggestion in order to improve the manuscript and also some errors have been detected

Detailed comments:

Abstract

The abstract lacks an introductory phrase that justifies the objective of the study. The experimental design is not explained either.

L10: Deleted "carried out in northern Poland and not skewed by pesticide use as it was carried out on an organic farm" and include after "organic rye field (L14 )" in northern Poland"

L12-14: indicate that the objective has also been to measure the yield and damage of the crop.

Keywords: "wildflower strips” and “functional diversity” are already in the Title of the manuscript and therefore should no longer appear in the keywords.

Introduction

The introduction does not reflect very well the previous knowledge about WFS and what is the contribution of this study to expand it. Various articles are cited, but the conclusions drawn from them are not conclusive. I recommend going deeper into the content of the paragraphs and giving more emphasis to the great work that has been done to study so many functional groups

L34: Change the two points to one point.

L40-44: include references to the role of WFS to improve the presence of pollinators.

L48-49: explain better the drawbacks of annual vs perennial WFS.

L54-57: What can be concluded from these articles? Are there any questions that still need to be answered? It is not clear why you cite all these studies, and the connection with your work is missing.

L63-65: I recommend including a phrase that highlights that you have evaluated the effect of the WFS considering the distance to the WFS.

L65-68. This sentence is very long and not very clear. The strong point of this article is that you have measured many functional groups of insects and also spiders. I would emphasize it more.

Material and methods

L75: add that the sampling has been done in one year.

L77: in one WFS? but the samplings have been done in the 2 WFS.

L81: "they were 300 x 6 and 200 x 6 m" these values do not correspond to the values in Fig 1.

L83: "Fagorpyrum esculemtum (59.0)" I would include the percentage used (40%) then you already explain that it is a variation of the Swiss mix.

L94: the data of F. esculentum (0.5 g / m2) does not correspond to 40% of the mixture, it is approximately 62%, Are total 0.5 + 0.3 g / m2?  Review this data.

L95-99: Delete this part, and include it in the results section.

L100-158: Separate into two different sections: 2.2. Experimental design and 2.3 Field survey method. Start the second section with L109.

L100: Figure reference should be in the text not in the title of the section.

L101: add "(A and B)" after "sectors".

L111-112: Include this justification when explaining the corresponding methodology (L155-158).

L119-120: this clarification is not necessary.

L122: change "4- 51-5%" by "4- 51-75%".

L135-139. It is not very clear. Did you walk through each section or in a single transect by distance? Clarify.

L124-129. Information is lacking on how the identification of insects and spiders has been made. In the case of the carabids, were all the individuals collected identified?. Regarding the plant-dwelling predatory insects, surely some insect pests have been found. How has this data been processed? For example, in the case of syrphids, some species, their larvae are predators, and the adults pollinators, how have they been taken into account?. In the same way, in pest insects sampling, predatory insects have probably been found, have they been eliminated from the results?

L146-150: How often?

L159-166: This should be included in Results not in M&M.

L168-171: delete this part, It has already been explained in the methodology.

L174-177: include that the species richness has also been calculated at the different distances from the crop. Has it not also been calculated using the Chao 2 estimator? Why? It is not clear how these results have been analyzed. Also with a GLZM?

L192: Why was the WFS not included in the analysis as distance 0?

L193: Why is the sector used as a fixed factor? I would include it as a random factor. Or it would justify better why it is interesting to evaluate the differences between sectors.

Results

L199-200: This should be explained in the statistical analysis section not here.

L204. Table S1: The title of the figure should be self-explanatory and reference to the methodology should not be made, the coverage values that represent the numbers from 5 to 0.1 should be included. In the same way for the rest of the figures where the same reference is indicated.

L206: Fig 2. I would add this figure in supplementary materials, it is a justification of the dependent variable used, in fact, these results are not mentioned in the discussion. However, in the discussion, the differences in floral coverage and richness between sectors are cited and these results could be presented in a Figure.

L208-2012: This analysis should also be explained in the statistical analysis section.

L233-234: What test has been used to see the differences? need to clarify.

L236: The results of Table S7 should be included in the manuscript, not in Supplementary materials. They are the most important analyzes of the article. Also, in Fig 3,4,5 these results could be included and explain the statistics in the results.

L248-253: The title of the figure should be checked, it refers to table 3. Is it referring to Table S7?

L259. There is an error: in table S7 the effect of the DIST has been statistically significant in the group of pollinators, but not in the predators.

L259-260: "There were two significant (P<0.05) or marginally significant (P<0.1) SECT × DIST interactions (Table S7)". Of what? complete the sentence.

L264-266: A post hoc analysis could be done to verify the differences between the different distances.

L277-279: Write all scientific names in italics.

L280-282: Add a statistical analysis on this, and specify it in the methodology.

L284-285: This phrase should be included in the discussion, not in results.

L294-309: Again the statistical analysis is not explained in the methodology.

Discussion

L322-337. In this paragraph several articles that evaluate the effect of distance to the WFS are cited, however, they are not correctly compared with the results obtained in this study. Does only the use of pesticides explain the differences or similarities with the different articles? It should be discussed why in WFS there is more richness and abundance of species than in the crop and why these parameters decrease with distance.

L333:"The only results....". It is not the only article that has observed a positive effect on bee abundance (eg. https://doi.org/10.1111/1365-2664.12479, https://doi.org/10.1111/1365-2664.12257), a better literature review should be done.

L355-362: And the effect of floral coverage? is it greater in one sector than in another? Differences between sectors have been found in arthropod abundance but not in richness. So this part should discuss this parameter.

L363-372. Long-term effect should be also consider, because in some studies, effects have been detected in the years following to the strip establishment (eg: https://doi.org/10.1111/1365-2664.12257)

Author Response

Dear Reviewer,

We appreciate all the comments and suggestions proposed, and we thank for your effort for improving the paper. Some remarks allowed us to avoid some important weaknesses or mistakes (e.g. comments on Syrphiidae). Below, we response to your comments in details and we list all the amendments we have done in the manuscript. The line numbers in the Reviewer’s comments refer to original submission.

Kind regards, on behalf of all authors,

Responses to comments

  • The abstract lacks an introductory phrase that justifies the objective of the study. The experimental design is not explained either.
    We have added an introductory phrase (L 12-13) and short description of the study design (L 19-21).
  • L 10: Deleted "carried out in northern Poland and not skewed by pesticide use as it was carried out on an organic farm" and include after "organic rye field (L14 )" in northern Poland".
    We have deleted words about N Poland, and we added the sentence about pesticides in the fragment about the study design (L 21).
  • L 12-14: Indicate that the objective has also been to measure the yield and damage of the crop
    Two aims (3, 4) about damage and yield have been added (L 18-19).
  • "Wildflower strips” and “functional diversity” are already in the Title of the manuscript and therefore should no longer appear in the keywords
    The key words listed above are deleted, and the two others are added: “ecological intensification” and “life traits diversity” (L 32-33).
  • The introduction does not reflect very well the previous knowledge about WFS and what is the contribution of this study to expand it. Various articles are cited, but the conclusions drawn from them are not conclusive. I recommend going deeper into the content of the paragraphs and giving more emphasis to the great work that has been done to study so many functional groups
    In line with the comment and recommendation we have added several sentences about cited papers (L 67-70, 71-73, 79-81). Also we underline assessing effect of distance from WFS (L 84-85), and multi-taxa/group approach in our study (L 87-88).
  • L 34: Change the two points to one point.
    The colon is replaced with comma, and the sentence is reworded (L 43)
  • L 40-44: Include references to the role of WFS to improve the presence of pollinators.
    The pollinators and the reference [8] are added to other groups of arthropods, which benefits from the WFSs (L51).
  • L 48-49: Explain better the drawbacks of annual vs perennial WFS.
    We added the information about the effect of the WFS’ age (L59-60), and about the role of perennial WFSs as habitat for overwintering (L 58-59 and 71-73).
  • L 54-57: What can be concluded from these articles? Are there any questions that still need to be answered? It is not clear why you cite all these studies, and the connection with your work is missing.
    Short summary of these results is added (L 67-73 and 79-80), and the difficulties in generalization of the study on the effect of distance from the WFSs was mentioned (L 79-81).
  • L 63-65: I recommend including a phrase that highlights that you have evaluated the effect of the WFS considering the distance to the WFS.
    The sentence “A key part of the study design was assessing the effect of distance from the WFSs.” is added (L 84-85).
  • L 65-68. This sentence is very long and not very clear. The strong point of this article is that you have measured many functional groups of insects and also spiders. I would emphasize it more

The sentence is split into two shorter sentences, and the information about the number of insect and spider groups is given (L 85-87).

  • L 75: Add that the sampling has been done in one year.
    Added (L 98)
  • L 77: in one WFS? but the samplings have been done in the 2 WFS
    “per sector” is added in L 102.
  • L 81: "they were 300 x 6 and 200 x 6 m" these values do not correspond to the values in Fig 1.
    Yes, because the samples were only collected in parts of the WFSs. This information is now added in the Section 2.2. (L 139-140).
  • L 94: the data of F. esculentum (0.5 g / m2) does not correspond to 40% of the mixture, it is approximately 62%, Are total 0.5 + 0.3 g / m2?  Review this data.
    All the values are verified, and we decided to add sawing density for the individual species. 40 % at F. esculentum was about the reduction of seed mass in relation to Swiss seed mixture. To explain it more, we added information on the changes in seed amount in g/100m2) (L 118)
  • L 100-158 Separate into two different sections: 2.2. Experimental design and 2.3 Field survey method. Start the second section with L 109.
    Corrected in line with the recommendation. Section 2.2 – L 130-143, Section 2.3 – L 145-198
  • L 100: Figure reference should be in the text not in the title of the section.
    The reference to Figure 1 is in the text now (L 131, 162 etc.)
  • L 101: add "(A and B)" after "sectors".
    Added (L 131).
  • L 111-112: Include this justification when explaining the corresponding methodology (L 155-158).
    The justification is moved to the paragraph about soil (L 194-195)
  • L 119-120: this clarification is not necessary.
    Deleted (L 154-155).
  • L122: change "4- 51-5%" by "4- 51-75%".
    Corrected (L157).
  • L 124-129. Information is lacking on how the identification of insects and spiders has been made. In the case of the carabids, were all the individuals collected identified?. Regarding the plant-dwelling predatory insects, surely some insect pests have been found. How has this data been processed? For example, in the case of syrphids, some species, their larvae are predators, and the adults pollinators, how have they been taken into account?. In the same way, in pest insects sampling, predatory insects have probably been found, have they been eliminated from the results?
    Thank you very much for this important comment.
    1) We have checked the datasets, and we have noticed, that Syrphiidae was mistakenly classified as predatory species, although we have collected the adults only. Therefore, we have deleted Syrphiidae from the datasets, and performed the analyses on “predators” again. It resulted in somewhat clearer and stronger relationships with Distance and Habitat (Table1).
    2) Pest insects were only collected with the use of method described in the point “E” (L 178-183)
    3) We have added the names of insect families, that were considered as predators or pollinators (L 170-172).
    4) The information about identification level is added (L 163-164).
  • L 135-139. It is not very clear. Did you walk through each section or in a single transect by distance?
    We clarified it by rewording the sentence (L 173-174)
  • L 146-150: How often?
    The date is given (1.07), which gives the information that assessment was done one time (L 184).
  • L 168-171: delete this part, It has already been explained in the methodology.
    Deleted (L 210-213)
  • Rev1: L174-177: include that the species richness has also been calculated at the different distances from the crop. Has it not also been calculated using the Chao 2 estimator? Why? It is not clear how these results have been analyzed. Also with a GLZM?
    Chao2, as index of total species richness, was only used for comparing WFSs and crop fields. Therefore it was estimated only for the distance of 45 m from WFSs, considered as a “control” transects that was not affected by the WFSs. From that reason it was not included to GLM, which covered mean number of species and abundances in sample sections or traps at all the distances (3, 6, 21, 45 m). We underline these difference in L 217-220. We also added the reference for Chao2 ([26], L 217)
  • L 192: Why was the WFS not included in the analysis as distance 0?
    We could not include WFS as the Distance = 0, as WFSs are different habitat when compared to crop field, which was analyzed in this part of analyses.
  • L 193: Why is the sector used as a fixed factor? I would include it as a random factor. Or it would justify better why it is interesting to evaluate the differences between sectors.
    Justification of such an approach is given in L 252-255.
  • Rev1: L 199-200: This should be explained in the statistical analysis section not here.
    The explanation is in the Section 2.3 (L 238-244).
  • L 95-99: Delete this part, and include it in the results section.
    Corrected: moved to the Results (L 277-282).
  • L 204. Table S1: The title of the figure should be self-explanatory and reference to the methodology should not be made, the coverage values that represent the numbers from 5 to 0.1 should be included. In the same way for the rest of the figures where the same reference is indicated.
    The explanation of the plant cover (and the scale used) is added. Unnecessary references to other parts the papers deleted, too (L 341, 401).
  • L 206: Fig 2. I would add this figure in supplementary materials, it is a justification of the dependent variable used, in fact, these results are not mentioned in the discussion. However, in the discussion, the differences in floral coverage and richness between sectors are cited and these results could be presented in a Figure.
    Indeed, the supplementary materials are better place for this correlation. We think however, that the verbal description of the differences in plant species richness between the Sectors and Habitats is enough as based on the mean values with CI. (L 288-291)
  • L 208-2012: This analysis should also be explained in the statistical analysis section.
    Deleted from the results, and explained in the Methods (L 239-240)
  • L 233-234: What test has been used to see the differences? need to clarify.
    Based on our best knowledge, there is no formal test that can be used for comparison just two Chao2 values. Therefore we assessed significance of the difference with the use of Chao2’s CIs. We added the explanation in the Section 2.3 (L 225-227).
  • L 236: The results of Table S7 should be included in the manuscript, not in Supplementary materials. They are the most important analyzes of the article. Also, in Fig 3,4,5 these results could be included and explain the statistics in the results.
    We agree with the suggestion. We inappropriately considered the results in the Table S7 as supplementary material. We moved the table to the main text as Table 1. The values in the column “Predators” are marked green, that indicates (the information to Reviewers)  that they are corrected (reason for it is explained in the point 22). However, we think that for Figures 3-5 (2-4 in corrected version), it is better to not combine with many numerical information on the GLM/GLZ results.
  • L 248-253: The title of the figure should be checked, it refers to table 3. Is it referring to Table S7?
    Yes, it was a mistake. Corrected, it is referring to Table 1 in re-submitted version (L 351-352).
  • L 259. There is an error: in table S7 the effect of the DIST has been statistically significant in the group of pollinators, but not in the predators.
    Indeed, it was a mistake. Corrected. Pollinators are added (L 358) however the “predators” remained as a marginally significantly related to the Distance.
  • L 259-260: "There were two significant (P<0.05) or marginally significant (P<0.1) SECT × DIST interactions (Table S7)". Of what? complete the sentence.
    We reworded the sentence. This is about the interactions between SECT and DIST in their relations with the spiders and insects (L 360)
  • L 264-266: A post hoc analysis could be done to verify the differences between the different distances.
    The variable DIST is numeric, continuous variable (see L 249). Therefore post-hoc analysis is not applicable here.
  • L 277-279: Write all scientific names in italics.
    Corrected
  • L 280-282: Add a statistical analysis on this, and specify it in the methodology.
    It was not tested. It is just information about the rate of increase in pest abundance. The explanation is added in Methods, section 2.3 (L 235-237).
  • L 284-285: This phrase should be included in the discussion, not in results.
    In the Discussion, there is a paragraph about the relationship between pests and predators (L 474-488).
  • L 294-309: Again the statistical analysis is not explained in the methodology.
    The sentence about the use of GLZ in analysis of ear damage is in Methods, section 2.3 (L. 240-241).
  • L 322-337. In this paragraph several articles that evaluate the effect of distance to the WFS are cited, however, they are not correctly compared with the results obtained in this study. Does only the use of pesticides explain the differences or similarities with the different articles? It should be discussed why in WFS there is more richness and abundance of species than in the crop and why these parameters decrease with distance.
    We added several sentences, which explain the results of other studies in more details, and also consider other factors, not only pesticides, when explaining the difficulties in generalization of results (L 466-469). The sentence about the differences in habitat structure between WFSs and crop fields, that explain the differences in animal diversity and abundance, is added, too (L 441 – 444).
  • L 333:"The only results....". It is not the only article that has observed a positive effect on bee abundance (eg. https://doi.org/10.1111/1365-2664.12479, https://doi.org/10.1111/1365-2664.12257), a better literature review should be done
    We thank the Reviewer 1 for recommending these papers. We used one of them (about blueberries) in the Discussion (L 460-465). However we were not able to refer to the second recommended paper as the direct analyses of the distance from the WFS are not presented in the paper.
  • L 355-362: And the effect of floral coverage? is it greater in one sector than in another? Differences between sectors have been found in arthropod abundance but not in richness. So this part should discuss this parameter.
    We would like to highlight two results of our study. Firstly, the differences between the Sectors concerned the mean number of arthropod species, too. We added the reference to Fig. 3 to provide an evidence for this conclusion. Secondly, as the plant cover was strongly correlated with the plant species richness, our results concerned these both vegetation variables. Therefore, we added the terms „number of species” and „vegetation cover” in L 492-496.
  • L 363-372. Long-term effect should be also consider, because in some studies, effects have been detected in the years following to the strip establishment (eg: https://doi.org/10.1111/1365-2664.12257)
    We agree, that this is important comment. We added the sentence in L 520-521.

Reviewer 2 Report

The authors describe a study of the impact of wildflower strips (WFS) on invertebrate diversity and abundance on transects into a rye fields.  The study site is free of pesticide use, which is frequently an unmanaged confounding factor in these types of studies. Although there are only two sample sites, and these within the same farm, the large number of invertebrate samples collected over the study season, and the temporal frequency of the sampling, make this a compelling study of the local impact of WFS on beneficial insects, including both pollinators and invertebrate predators of pest insects. Based on their findings the authors make practical suggestions for the introduction of WFS in agricultural landscapes in Poland.

Author Response

Dear Reviewer, 

we thank you for the effort and time spend for reviewing our paper, and for positive assessment. 

Kind regards, on behalf of all authors

Krzysztof Kujawa

Reviewer 3 Report

Dear Authors

I read with interest your manuscript “Annual wildflower strips as a tool for enhancing functional biodiversity in rye fields in an organic cultivation system”.

It is a manuscript that reports results on a field experiment conducted in untreated rye fields. Authors indagated many different arthropods indicators, proxy of biodiversity and pest pressure. There is a good presence of citations of previous works. The experimental design is clear. You performed a lot of samplings and procedures. Your results are rich in dataset and tables, provided as supplementary materials. The results not clearly support discussions and conclusions on one point, that I underlined. I also highlighted some other points to improve the manuscript before the publication.

- The first sentence of the Abstract is very long and complex to follow; thus, I recommend dividing it to make it easier to read.

- For the Keywords, I recommend using terms not present in the title in order to increase the search possibilities on browsers.

- L65-66 reads: “To compensate for the small number of replications (n = 2),”. Please evaluate whether to indicate that the test was done on one site only. The two replicas of the flower strip are in the same field!

- L81, add the year!

- Somewhere (materials and methods or in the discussion) it would be useful to describe the surrounding landscape, to allow the reader to understand the naturality around the field (e.g. presence of forests, cultivated fields, urban areas).

- L150, add the family name after the order name, e.g. (Heteroptera: Scutelleridae).

- For the "statistical analysis" part, you could indicate that the analyses were done with the software R and insert the relative citation. On L175 it is not clear whether the packet is of R or not (except by checking the reference n°24). On L187 it is clear.

- L215, please close the “ after “Methods.

- In section 3.4 I don't understand if in HAB =”field” all the data of the rye field have been merged or if you have only used the traps placed at 45 m from the WFSs, as I supposed reading L105-106. Please underline this aspect where you explain the two level of HAB.

- L277-279, the names of the species are not in italics. Moreover, I recommend adding also the order and family names when mentioned for the first time here.

- For figure 6 the x-axis seems categorical and not numerical as the previous figure appears. Please standardize.

- L299 Figure6 is mentioned, but Figure 6 represents something else. Please check!

- L322 and following read: "as the species richness and abundance (or activity density) of most of the species groups increased with decreasing distance from the WFS." But in the results, both in the figure and in the table, it seems that only a few categories respond to this behaviour and not many. Please insert more details in both results and discussion sections.

- It can be interesting if, in the discussion, the authors can discuss how arthropods can overwinter if the flowering strip are annual? Please advise for farmers and technicians who want to implement the technique in their field with this and other details. You claimed in the abstract: “should be taken into account in agri-environmental schemes in the Common Agricultural Policy after 2020.”

Author Response

Dear Reviewer,

We are grateful for the comments and for your effort for improving the paper. We consider all the remarks  and advices very helpful. Some of the remarks allowed us to avoid some important weaknesses or mistakes. Below, we response to your comments in details, and we list all the amendments we have done in the manuscript. The line numbers in the Reviewer’s comments refer to original submission.

Kind regards, on behalf of all authors

Responses to comments:

  • The first sentence of the Abstract is very long and complex to follow; thus, I recommend dividing it to make it easier to read.
    To make the sentence about the aim easier to read, we listed the detailed aims in the points 1-4.
  • For the Keywords, I recommend using terms not present in the title in order to increase the search possibilities on browsers.
    The key words listed above are deleted, and the two others are added: “ecological intensification” and “life traits diversity” (L 32-33).
  • L 65-66: “To compensate for the small number of replications (n = 2),”. Please evaluate whether to indicate that the test was done on one site only. The two replicas of the flower strip are in the same field!
    The information about the number of WFSs and fields studied is given now (L 86)
  • Somewhere (materials and methods or in the discussion) it would be useful to describe the surrounding landscape, to allow the reader to understand the naturality around the field (e.g. presence of forests, cultivated fields, urban areas).
    The information about the landscape structure is added in L 99-101.
  • L 81: Add the year!
    Added in L 98
  • L 150, add the family name after the order name, e.g. (Heteroptera: Scutelleridae).
    Added (L 188-189)
  • For the "statistical analysis" part, you could indicate that the analyses were done with the software R and insert the relative citation. On L175 it is not clear whether the packet is of R or not (except by checking the reference n°24). On L187 it is clear.
    We added the reference to the R environment in L 222.
  • In section 3.4 I don't understand if in HAB =”field” all the data of the rye field have been merged or if you have only used the traps placed at 45 m from the WFSs, as I supposed reading L105-106. Please underline this aspect where you explain the two level of HAB.
    Indeed, it was not clearly written. We use HAB=”field” only for the transects that were at the distance of 45 m from the WFSs, as an open field habitat presumably not affected by the WFSs. We explain this in L 247-248
  • L 277-279, the names of the species are not in italics. Moreover, I recommend adding also the order and family names when mentioned for the first time here.
    We followed this recommendation, changed font style to italic, and added order and family names
  • For figure 6 the x-axis seems categorical and not numerical as the previous figure appears. Please standardize.
    Indeed, we missed it. The x-axis has been corrected to numerical scale.
  • L 299 Figure6 is mentioned, but Figure 6 represents something else. Please check!
    It was a mistake. The reference to Fig 6 was unnecessary there. Deleted (L 413)
  • It can be interesting if, in the discussion, the authors can discuss how arthropods can overwinter if the flowering strip are annual? Please advise for farmers and technicians who want to implement the technique in their field with this and other details. You claimed in the abstract: “should be taken into account in agri-environmental schemes in the Common Agricultural Policy after 2020.”
    We added a sentence in L 517-521.
  • L 322 and following read: "as the species richness and abundance (or activity density) of most of the species groups increased with decreasing distance from the WFS." But in the results, both in the figure and in the table, it seems that only a few categories respond to this behaviour and not many. Please insert more details in both results and discussion sections.
    More details are given in L 446-447.